# Asking Multimodal Clarifying Questions in Mixed-Initiative Conversational Search

Submission Id: 1033

## ABSTRACT

In mixed-initiative conversational search systems, clarifying questions are used to help users who struggle to express their intentions in a single query. These questions help uncover user's information needs and resolve query ambiguities. We hypothesize that in scenarios where multimodal information is pertinent, the clarification process can be improved by using non-textual information. Therefore, we propose to add images to clarifying questions and formulate the novel task of asking multimodal clarifying questions in open-domain, mixed-initiative conversational search systems. To facilitate research into this task, we collect a dataset named Melon that contains over 4k multimodal clarifying questions, enriched with over 14k images. We also propose a multimodal query clarification model named Marto and adopt a prompt-based, generative fine-tuning strategy to perform the training of different stages with different prompts. Several analyses are conducted to understand the importance of multimodal contents during the query clarification phase. Experimental results indicate that the addition of images leads to significant improvements of up to 90% in retrieval performance when selecting the relevant images. Extensive analyses are also performed to show the superiority of Marto compared with discriminative baselines in terms of effectiveness and efficiency.

## CCS CONCEPTS

• **Information systems** → Evaluation of retrieval results.

## KEYWORDS

Query clarification, Multimodal query understanding

**ACM Reference Format:**
Anonymous Author(s). 2024. Asking Multimodal Clarifying Questions in Mixed-Initiative Conversational Search. In *Proceedings of the ACM Web Conference 2024 (WWW '24), May 13-17, 2024, Singapore.* ACM, New York, NY, USA, 12 pages. https://doi.org/10.1145/XXXXXXX.XXXXXXX

## 1 INTRODUCTION

Traditional search systems often struggle to provide relevant results that meet a user's information need when they encounter incomplete or ambiguous queries. Query clarification has emerged as a promising approach to address this challenge [4, 72]. It enables systems to interact with users to clarify their information needs before presenting search results [26, 51]. While previous research has focused primarily on unimodal interactions, the growing

popularity of smart displays underscores the increasing demand for multimodal communication [36]. Specifically, the use of multimodal information improves effectiveness in various information retrieval (IR) tasks, including image-based search [68], e-commerce cross-modal retrieval [25], and fashion recommendation [70]. In this context, multimodal information not only provides a visually appealing experience to the users but also enhances both the system and user performance, allowing for the integration of complementary information across multiple modalities.

**Motivation.** As the interest in multimodal information-seeking conversations continues to grow [23, 62], we study the use of multimodal information in query clarification. We hypothesize that the addition of relevant images to clarifying questions within a conversational context can enhance the user experience and performance, providing a rich source of information that augments textual input. Consequently, this should lead to increased awareness and understanding of the information need and various aspects of it such as domain knowledge. Fig. 1 shows two example conversations — without and with images. We see that when a user inputs the query

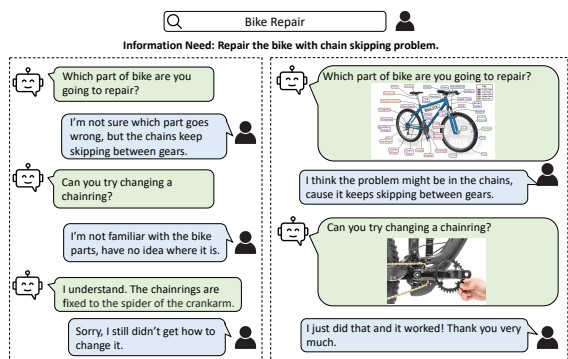

**Figure 1: An example of incorporating multimodal information into the query clarification phase.**

"bike repair" with the underlying information need of "repair a bike with a chain skipping problem," presenting a clarifying question accompanied by an image of different parts of a bike can provide a visual representation of the intended query, allowing the user to see the object they are searching for. The potential benefits of this approach are manifold, including: (i) displaying the appropriate set of images with clarifying questions reassures the user that the system has correctly understood their request; (ii) showing images can help the user acquire a more detailed and complete comprehension of the search topic domain; and (iii) it could enable the user to engage in discussions and inquiries, pertaining to visual content, such as a specific part of a bike. Ultimately, these benefits help ensure that the retrieved results are better aligned with the user's actual information needs, leading to improved retrieval effectiveness and efficiency. Besides, as visual content provides a more engaging and

immersive interaction, it leads to increased user engagement [37] and satisfaction with the search experience [23]. E.g., users may have difficulty understanding the conversation when the system relies on text descriptions to explain the process of a bike repair.

**Research goals**. In this study, we formulate a novel task of asking multimodal clarifying questions in open-domain mixed-initiative conversational search systems. We define multimodal query clarification (MQC) as *asking clarifying questions enriched with images in a text-dominant conversation* and present the workflow depicted in Fig. 2. Initially, when a user submits a search query with some intention that is concealed from the system, the system retrieves relevant documents and evaluates their appropriateness for presentation to the user. For ambiguous queries for which the system struggles to understand the user's intent, a clarifying question is asked. Our approach employs a classifier to determine whether to include images in the clarifying question. Upon confirmation (this type of question is defined as a **multimodal clarifying question**), images are retrieved and displayed to the user with the clarifying question. In this case, visual information serves as a valuable resource, enhancing user comprehension and supporting informed responses. Therefore, combined with prior context, it can deliver precise and relevant retrieval results.

To advance research into the proposed task, following the workflow described above, we collect a new dataset *Melon* (**m**ultimodal qu**e**ry c**l**arification c**on**versations), based on the text-only clarifying question dataset ClariQ [3]. Melon contains over 4k multimodal clarifying questions, which are enriched with over 14k images. With Melon, we investigate the following research questions: **RQ1**: *What is the effect of including multimodal content in the document retrieval task?* To address **RQ1**, inspired by generative retrieval techniques in the text-only retrieval domain [11, 31], we propose Marto (**m**ultimod**a**l que**r**y clarifica**t**ion m**o**del) and compare it with several state-of-the-art unimodal models. Marto is based on a multimodal generative framework and adopts a multi-task fine-tuning approach to train different stages of Marto with different prompts. We initially train Marto to generate 'true' or 'false' labels to assess if a given clarifying question is multimodal. For 'true' cases, it selects an image based on its similarity to the question. We also develop a generative document retrieval method where all questions with or without images are trained to generate document IDs. By leveraging pre-trained knowledge, this approach equips Marto with enhanced retrieval ability in an end-to-end manner. Our analyses indicate that adding images can lead to up to 90% improvements in performance.

In **RQ2**, we address: *How does the attachment of different images affect the performance of multimodal query clarification in terms of retrieval accuracy?* We answer this question by conducting a performance comparison between Marto and its variants by attaching different images and several multimodal baselines. We show that selecting the relevant image is beneficial to the retrieval performance compared with attaching a random one. Our findings provide important insights into the design of Marto and aid the development of more advanced conversational search systems that take advantage of the rich information in multimodal interactions.

To further assess the performance of Marto and address **RQ3**: *Are generative clarification models more effective and efficient for document retrieval?*, we conduct experiments and analyze the training progress in terms of loss and validation of competitive generative and discriminative clarification models. Our results reveal that Marto, in comparison to its discriminative counterpart VisualBERT, exhibits considerably reduced training time (0.77 vs. 8.26) and inference time (0.67 vs. 1.13). Thus indicating the superiority of generative clarification models in document retrieval .

Finally, to understand the significance of multimodal content in the query clarification process and its potential impact on users, we address **RQ4**: *How does the inclusion of multimodal clarifying questions impact the user response?* To answer this question, we perform analyses from both dataset and model perspectives. Our findings show that using images during the clarification phase leads to more contextualized answers. This results in richer semantic information compared to datasets that only feature text-based clarifications.

**Contributions**. Our main contributions are as follows:
- We define a novel task of MQC in a mixed-initiative conversational search system, which adds image information in clarifying questions to improve the downstream document retrieval task.
- To facilitate the offline evaluation of the task, we benchmark a large-scale dataset called Melon and propose Marto for representation learning of multimodal clarifying questions.[1]
- Extensive analyses are performed to explore the impact of multimodal clarifying questions on user response and their role in improving retrieval performance.

## 2 RELATED WORK

**Query clarification in mixed-initiative search systems**. Query clarification refers to the process of improving a search query by adding more context or details to it. In recent years, query clarification has become an essential task in various domains, such as entity disambiguation [21], voice [29], dialogue [8, 48], question answering [9, 51, 69], recommendation [18, 76]. In mixed-initiative search systems where the initiative shifts back and forth between agents and users [5, 28], asking clarifying questions has received considerable attention [27, 55, 74]. Efforts have been made to investigate the role of clarifying questions in mixed-initiative systems, recognizing their potential to improve search quality and user experience [1, 40, 63]. To explore when to ask clarification questions, the TREC CAsT 2022 track included a task where the system can either take the initiative by posing questions or generating a response.[2] Resources have been proposed to facilitate the offline evaluation of such systems. For example, MIMICS is a large-scale dataset sampled from the Bing query logs [59, 73]. In open-domain information-seeking conversations, Qulac is a clarification dataset that contains clarifying questions and answers [4]. ClariQ expands Qulac to a multi-turn format and has a larger scale [3]. Despite the progress in mixed-initiative search clarification, there is a significant gap in our understanding of how to incorporate multimodal information in query clarification.

**Multimodal IR**. Multimodal IR involves integrating multimodal query processing techniques to effectively capture users' diverse

---

[1]A link to the data and code repository will be added upon acceptance. To preserve the copyright of the images we will only share the public URLs to the images. Also, we will ask the researchers to sign a license agreement.
[2]https://www.treccast.ai/

search intent [7, 41, 43, 56], whose techniques can be applied in various IR scenarios [22, 25, 54, 58, 68, 70]. In mixed-initiative systems, multimodal content is leveraged to enhance retrieval efficiency [23, 25, 79] and improve user experience [38, 42]. Inspired by the success of generative LLMs [10, 50], a set of multimodal pre-trained generative models have been proposed recently [17, 35, 75, 77]. Subsequent efforts have proven their effectiveness in IR tasks such as query reformulation [71], question answering [12, 57], cross-modal retrieval [49]. VLT5 [17] is the state-of-the-art and shows strong performance, especially in the image caption task. We incorporate images into the clarification phase of mixed-initiative conversations and develop our model using VLT5 as the base model.

**Generative IR**. Generative retrieval, unlike traditional IR systems that follow an "index-retrieve-then-rank" pipeline, generates identifiers of relevant documents in an end-to-end way [13, 15, 16]. This concept was initially introduced in the context of entity linking tasks [11], where autoregressive models were employed to generate entity names, resulting in improved accuracy for this task. Subsequent work has explored the generation of document titles for fact-checking [14], recommendation tasks [64]. Efforts have also been made to investigate and compare the effect of different document identifiers [13, 60]. E.g., Bevilacqua et al. [6] leverage n-grams as possible identifiers. Wang et al. [65] propose generating the document id and improve the system with a prefix-aware decoder. Lee et al. [31] use a contextualized embedding matrix to generate the target sequence.

Motivated by the success of LLMs and multimodal pre-trained generative models [17, 44, 49], we propose Marto, the first multimodal generative retrieval model that takes multimodal information as input and generates the corresponding document identifier.

## 3 TASK DEFINITION AND WORKFLOW

In a search system, the user submits a query with a latent intention or information need. The system's goal is to accurately predict this intention and retrieve relevant results. To achieve this, the system can use multimodal clarification questions, which are meant to help capture the user's needs. We define the multimodal query clarification (MQC) task as follows: *multimodal query clarification refers to scenarios where the user-system interactions are mainly in text, while the clarifying questions are potentially enriched with multiple images.* The main assumption is that the images (i) are related to the conversation topic and the posed clarifying question, and (ii) provide further contextual information on the topic. E.g., in Fig. 1, if a user is not familiar with different parts of a bike, attaching an image to the clarifying question can aid in their comprehension and provide additional knowledge on the topic. Under this case, multimodal clarifying questions help enhance user performance, as they offer a richer context and enable users to respond more effectively with accurate answers. While answers to both multimodal and unimodal clarifying questions are textual, we hypothesize that answers provided in a multimodal setting are more valuable, due to the richer semantic information conveyed by the inclusion of images, ultimately resulting in improved document retrieval performance.

Inspired by [4], we propose a workflow for MQC as shown in Fig. 2. The whole process begins when (i) a user submits a query, the system then predicts the information need by converting the

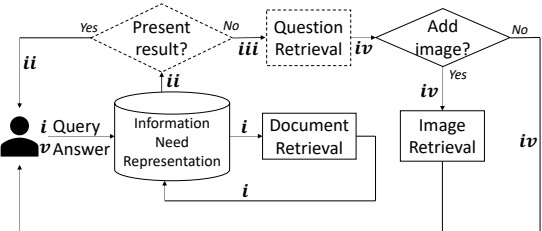

**Figure 2: A workflow of adding MQC phase in a conversational search system. Hashed modules remain the same as in the unimodal clarification system presented in [4].**

query into an information need representation, and passes it to the document retrieval module to retrieve a ranked list of documents; (ii) at this point, the system decides whether to present the result to the user or ask a clarifying question, based on the confidence score assigned to the retrieved documents – i.e., if the query is ambiguous or the information need is not clear, the confidence score falls below a threshold; (iii) in such cases, the system selects or generates a clarifying question to ask the user; (iv) after that, the system determines whether the question requires an image or not; if an image is needed, the image selection module returns an image to be added to the question. Otherwise, the question is presented to the user without any images; (v) according to the clarifying questions and the corresponding images (if any), the user provides an answer. The system repeats this procedure until it reaches a high confidence score or the maximum number of questions are reached, and finally presents the relevant documents to the user.

## 4 DATA COLLECTION & ANALYSIS

To facilitate MQC research, we describe how we create the dataset Melon. We then perform analyses on Melon to answer **RQ4**[3].

### 4.1 Data collection

Our approach to constructing Melon draws inspiration from existing unimodal clarification datasets [2–4]. In Melon, initial user queries can be subdivided into multiple facets that reflect the user's real intention. A set of clarifying questions are associated with every facet of the query. Afterwards, user responses are collected to answer each clarifying question and provide insight into the corresponding facet. We first leverage topics from the TREC Web Track 2009−2012 [19] as user queries and reuse the collected subtopics as facets. To create a more enriched and diverse dataset that combines the existing resources with new instances, we utilize the clarifying questions from the text-only ClariQ dataset [3] while also collecting new ones from scratch. We then enrich all the clarifying questions with images and add new answers to enhance the dataset's utility.

**Data collection pipeline**. We implement our data collection pipeline in 3 phases. (**Phase 1**) collect clarifying questions tailored to be multimodal from both ClariQ and from scratch; (**Phase 2**) collect a diverse set of images that can be attached to the collected clarifying questions; and (**Phase 3**) collect new answers for the clarifying questions presented with their corresponding images.

**Phase 1: Collecting multimodal clarifying questions**. The collection of suitable clarifying questions, specifically those that

---

[3]The ethics statement of Melon is listed in Appendix D.

pertain to image attachment, is a crucial step in creating a high-quality MQC dataset. We gather these questions from two distinct sources: (i) existing ClariQ questions; and (ii) newly collected clarifying questions. To obtain the first set of questions (which we call **set 1 questions**), we employ two expert annotators to classify the existing ClariQ questions as either **multimodal** (i.e., questions suitable for image attachments) or **unimodal** (i.e., questions not appropriate for image attachments).[4] The annotators initially agree on 95% of the annotated questions with Cohen's inter-annotators' agreement indicating a strong level of agreement ($\kappa = 0.82$). In case of disagreement, we resolve the conflict by asking the annotators to discuss and reach a final decision. For the second set of questions (**set 2 questions**), we design a human intelligence task (HIT) on Amazon Mechanical Turk (AMT)[5] where we ask annotators to generate new multimodal questions given the user query and ensure that the user facet remains undisclosed to them throughout the process. We provide detailed instructions on how to generate these questions and urge our annotators to follow these steps to create high-quality multimodal clarifying questions: (i) enter the user query into an image search engine (e.g., Google image search) and scan the first three pages; (ii) scan the image-oriented question suggestions at the top of the result page, which provide useful hints for image-oriented aspects of the query; (iii) check the query auto-complete suggestions for additional hints; and (iv) write three questions, focusing on different facets of the query.

**Phase 2: Image collection/attachment**. In Phase 2, we aim to associate each clarifying question obtained in Phase 1 with relevant images. To achieve this, we instruct expert annotators to use a search engine of their choice and find images that meet the requirements specified for each clarifying question. For instance, in the case of Fig. 1, the annotator is guided to search for "bike diagram" in the image search engine, using both the query and clarifying question as a reference. Annotators are then required to select up to three images that they find most relevant to the clarifying question. To ensure diversity, we require that each selected image depicts a distinct aspect related to the query-clarifying question combination, and that annotators record the corresponding aspect of each image. E.g., annotators may provide the images of different types of bikes, such as 'mountain bike diagram', 'city bike diagram.'

**Phase 3: Answer collection**. In Phase 3, we create a new AMT HIT to collect answers for the multimodal clarifying questions. To ensure reliability of our results, we gather new answers for all questions instead of relying on the existing answer set from ClariQ. Our hypothesis is that the inclusion of images can have a significant impact on users' behavior, leading to inaccuracies in the text-only ClariQ answers. We provide the annotators with the original user query, topic facet, and multimodal clarifying questions along with their corresponding images. We instruct them to imagine as users being part of an ongoing conversation with the facet as their real intention. We encourage them to provide natural language responses as if they were engaging in a dialogue with the system. We emphasize the importance of considering both the question and accompanying images when providing answers. Our goal is to obtain accurate and informative answers that would help improve

---

[4]We recruit the annotators from Appen (https://appen.com/).
[5]https://mturk.com

**Table 1: Statistics of the Melon dataset.**

| | |
|---|---|
| # topics | 298 |
| # facets | 1,070 |
| # all questions | 4,969 |
| # set 1 questions | 3,365 |
| # set 2 questions | 1,604 |
| Avg. question per topic (std.) | 16.67 (3.59) |
| Avg. # terms per question (std.): | 9.85 (2.63) |
| # images | 14,869 |
| Avg. # images per question (std.) | 2.99 (0.12) |
| # answers | 18,533 |
| Avg. # answers per question (std.) | 3.73 (1.61) |

**Table 2: Answer comparison between Melon and ClariQ.**

| | Avg. terms (std.) | Mid. terms | Max. terms | Yes/no answers (%) | Vocab. size |
|---|---|---|---|---|---|
| ClariQ | 8.12 (4.58) | 9 | 30 | 10.25 | 4,561 |
| Melon | 10.76 (4.73) | 10 | 96 | 3.06 | 8,622 |

the performance of the MQC system. We also adopt several quality control methods to ensure the quality of Melon; cf. Appendix A.

### 4.2 Data analysis

**Dataset statistics**. Table 1 provides an overview of the basic statistics of Melon. The dataset comprises a total of 298 search topics and 1070 facets. It consists of 4,969 multimodal clarifying questions accompanied by 14,869 associated images, resulting in an average of 2.99 images per question. Among these questions, 67.7% originate from ClariQ, whereas 32.3% are newly collected. Additionally, the dataset includes 18,533 answers, with an average of 3.73 answers per question, equating to one answer per facet-question pair. To have a more comprehensive understanding of Melon, we then compare the language usage in ClariQ questions (**set 1 questions**) and AMT questions (**set 2 questions**). Due to page limitations, details are included in Appendix B.

**Answer characteristics**. To address **RQ4** *from a dataset perspective*, we analyze the differences in user responses between unimodal and multimodal conversational systems. We compare the user responses collected in **Phase 3** with the original responses of ClariQ — responses to unimodal questions. Table 2 lists various characteristics of the answer collection of ClariQ and Melon. Additionally, Figure 3 illustrates the distribution of answer lengths in the two datasets. When examining Table 2 and Figure 3, we observe notable differences between the two datasets. The responses in Melon exhibit more engagement, resulting in longer answers (with an average of 10.76 terms) compared to ClariQ (with an average of 8.12 terms). Furthermore, there is a lower percentage of one-word answers such as yes/no (3% in Melon vs. 10% in ClariQ), indicating that the presence of multimodal clarifying questions encourages users to provide more detailed and informative responses. Moreover, the vocabulary size of Melon is nearly twice as large as that of ClariQ. This suggests that the inclusion of multimodal clarifying questions enables users to provide responses with richer semantic information, resulting in a more diverse and expanded vocabulary.

**Summary**. We show that text-only clarifying questions can be enriched with images to create a multimodal dataset. We propose

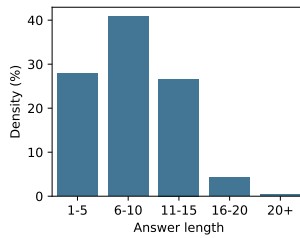 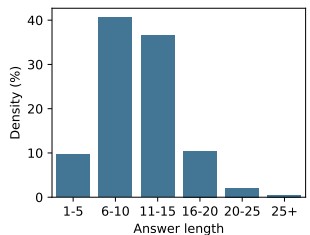

**(a) Unimodal answers**    **(b) Multimodal answers**

**Figure 3: Distribution of answer length in (a) unimodal and (b) multimodal datasets. Density represents the proportion of each type of answer in the answer set.**

Melon for offline evaluation of MQC models. Our analysis on the ClariQ and Melon answers suggests that the inclusion of images in clarifying questions yields more comprehensive user responses; see Section 6.1 for more on this in the context of **RQ4**.

## 5 MULTIMODAL QUERY CLARIFICATION

**Problem formulation**. Similar to Qulac [4] and ClariQ [3], given a list of topics denoted as $T = \{t_1, t_2, \ldots, t_k\}$, each topic is associated with a set of facets $\Gamma = \{F_1, F_2, \ldots, F_k\}$, where $F_i = \{f_i^1, f_i^2, \ldots, f_i^{n_i}\}$ represents the corresponding facets to the $i$-th topic $t_i$. $n_i$ denotes the number of facets of topic $t_i$. In addition, $Q_i = \{q_i^1, q_i^2, \ldots, q_i^{l_i}\}$ denotes the set of $l_i$ clarifying questions that belong to topic $t_i$. Our approach is different from Qulac in the sense that each clarifying question $q_i^j$ is associated with a tag $b_i^j \in \{0, 1\}$ representing if the question is multimodal, i.e., has images, or not. If $b_i^j = 1$, $I_i^j$ is the image set of question $q_i^j$. Furthermore, for each topic $t$, facet $f$, question $q$, and the associated question images $I$ (if any), we define an answer function $A(t, f, q, I) \rightarrow a$, which maps the current conversation context to a user answer. Following [2], we borrow all the topic set $T$, facet set $\Gamma$, and relevance assessments from TREC Web Track 2009–2012. As described in Section 4.1, we manually collect the question set $Q$ (**Phase 1**), the image set for each question $I$ (**Phase 2**), and the answers $A$ (**Phase 3**).

**Marto architecture**. Fig. 4 and 5 illustrate the overall architecture of the model we propose for MQC task, Marto. Our approach builds on the workflow described in Section 3 and consists of several modules. Following the workflow in Fig. 2, when a conversational system enters question the clarification phase after a user query submission (phase i, ii), a clarification question is retrieved (**clarification question retrieval module**). In phase iii, a question classification module is employed to judge if the question is multimodal (**multimodal-enhanced question classification module**). If yes, several images are selected (**image selection module**) in phase iv. In phase v, the users provide the response before all the additional information is fed into the document retrieval module to retrieve documents (**document retrieval module**). In Marto, we focus on the last three modules shown in Fig. 4, as the other modules can be adopted from existing unimodal clarification systems [4]. As depicted in Fig. 5, Marto is based on a multimodal generation model named VLT5 [17]. We develop a multitask fine-tuning strategy and use a single model with different prompts to fine-tune different subtasks [32, 44]. The subtasks are detailed below.

**Multimodal-enhanced question classification**. While all questions in the Melon dataset are assumed as multimodal by human

annotators, our preliminary experiments have shown that attaching images to some questions can introduce additional noise and adversely affect the retrieval performance. We denote these questions as the text-enhanced questions (**TEQ**). However, for some other questions, adding images has a positive effect from the model's perspective. We denote them as the visually-enhanced questions (**VEQ**). To classify questions as either TEQ or VEQ, multimodal-enhanced question classification involves training a binary classifier that labels the questions from the model prediction as suitable for image association or not. Section 6.1 offers an analysis of the reasons behind the discrepancy between the predictions made by the model and those made by humans.

We train our classification model based on a multimodal generative model named VLT5 [17]. VLT5 is a state-of-art multimodal pretrained model that takes text and images as input and generates text as output. The text input of our model includes three parts: task prefix, user query, and clarifying question. The task prefix is a short prompt that differentiates this task from others. We use "question classification:" as the task prefix. The user query and the clarifying questions are appended and separated by a special token [SEP]. All the text input is then tokenized and embedded before being passed to the encoder. Following [50], we incorporate relative position bias to each self-attention layer. As a result, the text input is represented as $e^t = \phi_T(\langle p_1 \rangle, t, q)$, where $\phi_T$ represents the text-embedding function and $\langle p_1 \rangle$ is the short prompt. Due to the lack of image input for this task (we only classify text-only questions as VEQ/TEQ), we mask the image domain of the model. We train the model to generate the labels directly. For VEQs, it generates "true", and for TEQs, it generates "false." This can be represented as:

$$y_q = \text{VLT5}(\phi_T(\langle p_1 \rangle, t, q)), \tag{1}$$

where $y_q$ is the label of each question.

**Weakly supervised label generation**. We propose a weakly supervised label generation method to automatically generate ground-truth VEQ/TEQ labels. We train two document retrieval models, where the first one takes a text-only (unimodal) input (e.g., BERT), called TOR, and the second is a multimodal version (e.g., Visual-BERT), called MUR. To make a fair comparison, TOR and MUR share the same basic model structure and text input, including user query, clarifying question, and corresponding answer, where each is separated by the [SEP] token. For MUR, we additionally attach the corresponding images of the clarifying questions as the image input. We compare the retrieval performance of these two models and calculate the relative improvement $\Delta \text{nDCG} = \text{nDCG(MUR)} - \text{nDCG(TOR)}$. If MUR performs better ($\Delta \text{nDCG} > 0$), we assign a positive label to the clarifying question (VEQ). Otherwise, the question is labeled as negative (TEQ). In our experiments, we use the average of $nDCG@\{1, 3, 5\}$ as the nDCG score.

**Image selection**. This module aims to rank the most relevant images given a clarifying question. We propose a simple but effective cross-modal retrieval method that does not require training. Inspired by the success of pre-trained image-text models in the cross-modal retrieval task, we adopt the pre-trained CLIP [49] embedding to obtain the embedding of each clarifying question and image. We then rank the images based on their similarity scores to the question. The selected image $I$ for question $q$ given candidate

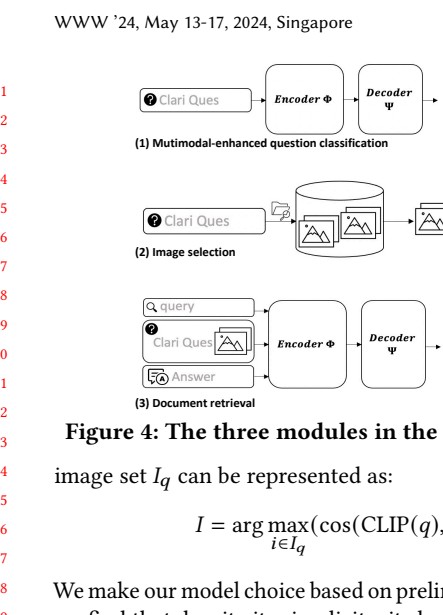

Figure 4: The three modules in the Marto model.

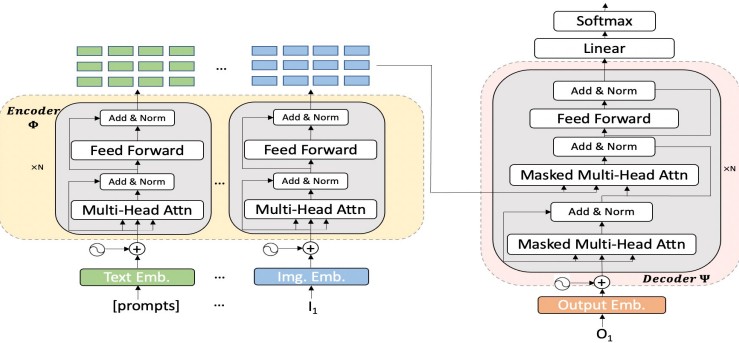

Figure 5: The detailed structure of Encoder $\Phi$ and Decoder $\Psi$.

image set $I_q$ can be represented as:

$$I = \arg\max_{i \in I_q}(\cos(\text{CLIP}(q), \text{CLIP}(i))) . \quad (2)$$

We make our model choice based on preliminary experiments where we find that despite its simplicity, it shows good performance.

**Document retrieval**. Our document retrieval module ranks the documents for a topic, a given user query, a VEQ clarifying question with images and the user answer. For the TEQs, we keep the same model structure but mask the image input. Our model adopts the idea of generative retrieval method [11, 15, 78], which has shown great research potential in the text-only retrieval domain under the recent success of generative NLP.

Similar to the multimodal-enhanced question classification module, this module utilizes the VLT5 model [17] for the document retrieval task. However, we employ a different prompt specifically for document retrieval, using "document retrieval:" as the task prefix. The rest of the text input contains the topic, the user query, the clarifying question, and the user answer, each separated by the [SEP] token. The tokenization and embedding methods are the same as the multimodal-enhanced question classification module. To obtain the embedding of image $I$, following previous works [17, 34, 35], we first detect several object regions denoted as region of interest (ROI). All ROIs are detected using the object detection model FasterRCNN [52] pre-trained on the Visual Genome dataset [30]. We then add the ROI features with ROI bounding box coordinates and region ids $\in \{1, 2, \ldots, n\}$ before fed into a linear layer. The final visual vector is represented as $e^v = \phi_V(I)$.

In the generative IR literature, the model directly generates identifiers of a set of contexts given an input query [11, 13, 15]. Following this approach, after feeding the image and text embedding into the VLT5 encoder, we aim to generate the sequence of identifiers of relevant documents. We extract the document keywords as the unique identifier of each document, for the reason that keyword represents the most important part of each document and has a natural language format. For each document, we take the top-5 keywords as the corresponding identifier. We train the decoder to generate the identifier of the top-5 ground-truth relevant pages, each concatenated by the [SEP] token. The training process can be represented as:

$$y_d = \text{VLT5}(\phi_T(\langle t_2 \rangle, t, q, a), \phi_V(I)) , \quad (3)$$

where $y_d$ is the generated document identifier sequence, $t_2$ is the prompt of the task, $a$ is the user answer.

For multimodal-enhanced classification and document retrieval, we adopt the standard generation loss when fine-tuning the VLT5. During inference, we use constrained beam search [11] to rank documents. This allows us to limit the generated content to be within the pre-defined candidate set with a generative score, i.e., the keyword identifiers of all documents in our corpus.

## 6 EXPERIMENTS

We experiment on the Melon dataset by comparing Marto with several state-of-the-art methods, including lexical methods (i.e., OriginalQuery, QL, BM25, LambdaMART), pipeline-based methods (i.e., BERT+CLIP+CLIP, BERT+CLIP+VLT5), methods under a multi-task framework (i.e., BERT, T5, VisualBERT, VisualBERT_w/o QC), and variants of Marto (i.e., Marto_w/o QC, Marto_VLP, Marto_random-image, Marto_trel-image, Marto_oracle-best-image). Appendix C details our experimental setup and baselines.

### 6.1 Results & Analyses

**Primary findings**. In Table 3 we report the performance of Marto and other several baselines. Our findings indicate that incorporating clarifying questions significantly improves the performance of document retrieval compared to directly using the query alone (i.e., OriginalQuery) consistent with findings in [4]. Moreover, large pre-trained language models like BERT demonstrate superior performance compared to lexical methods such as BM25. Furthermore, our analysis reveals that methods that adopt multitask learning with a more compact structure outperform pipeline-based methods (e.g., BERT+CLIP+VLT5), allowing for the sharing of feature representations across tasks. Detailed analyses are listed below.

**Image vs. no image (RQ1)**. Regarding the impact of images on the document retrieval task (as addressed above), we observe that clarification systems with multimodal information outperform those relying solely on text (e.g., BERT vs. VisualBERT) in Table 3. Also, we find that adding images can lead to up to 90% improvements in performance (Marto vs. T5). In terms of the convergence, as shown in Fig. 6, multimodal methods (square marker) take less time to reach good performance than text-only methods (circle marker), demonstrating the positive impact of incorporating multimodal information on model optimization and training time. Also, classifying the questions into VEQ and TEQ categories before directly performing the retrieval task can further boost the results. For VisualBERT, removing the multimodal-enhanced question classification module results in a 3.21% decrease in P@1 (VisualBERT vs. VisualBERT_w/o QC), while for Marto, P@1 is reduced by 13.14% (Marto

**Table 3: Experimental results of Marto compared with baselines. QC represents the multimodal-enhanced question classification module. Img. represents whether the model takes image as input or not. Numbers in bold represent the best-performing model. * denotes Marto shows significant improvement under the significance test. All the numbers are shown in %.**

| | Img. | MRR | P@1 | P@3 | P@5 | nDCG@1 | nDCG@3 | nDCG@5 | ERR@1 | ERR@3 | ERR@5 |
|---|---|---|---|---|---|---|---|---|---|---|---|
| OriginalQuery | ✗ | 14.06 | 18.75 | 14.58 | 11.88 | 5.05 | 3.64 | 3.77 | 2.44 | 2.83 | 3.28 |
| QL | ✗ | 14.71 | 17.64 | 19.61 | 20.00 | 12.35 | 12.49 | 12.09 | 4.96 | 6.41 | 8.08 |
| BM25 | ✗ | 20.31 | 23.44 | 24.48 | 24.69 | 11.59 | 10.07 | 9.98 | 4.59 | 7.05 | 7.19 |
| LambdaMART | ✗ | 24.39 | 23.95 | 25.41 | 24.78 | 12.32 | 11.89 | 11.39 | 4.70 | 9.02 | 9.62 |
| BERT+CLIP+CLIP | ✓ | 35.62 | 29.85 | 28.87 | 27.69 | 17.02 | 18.17 | 17.19 | 6.73 | 12.39 | 15.34 |
| BERT+CLIP+VLT5 | ✓ | 37.90 | 34.52 | 33.70 | 32.85 | 20.92 | 21.15 | 19.09 | 9.26 | 13.48 | 13.54 |
| BERT | ✗ | 26.43 | 25.00 | 26.56 | 28.24 | 13.17 | 13.70 | 14.52 | 5.76 | 9.16 | 10.77 |
| T5 | ✗ | 27.56 | 25.10 | 26.04 | 26.88 | 13.62 | 14.16 | 14.28 | 5.18 | 9.84 | 10.70 |
| VisualBERT | ✓ | 41.56 | 33.75 | 32.01 | 31.94 | 19.55 | 20.80 | 18.85 | 7.99 | 15.13 | 14.39 |
| VisualBERT_w/o QC | ✓ | 33.28 | 30.54 | 32.31 | 32.45 | 15.31 | 16.16 | 19.55 | 6.88 | 12.47 | 12.85 |
| Marto | ✓ | **54.70*** | **53.38*** | **40.47*** | **36.65*** | **30.66*** | **24.57*** | **23.81*** | **15.63*** | **20.21*** | **21.64*** |
| Marto_w/o QC | ✓ | 46.32 | 40.24 | 36.65 | 30.97 | 20.50 | 18.26 | 20.32 | 10.24 | 14.57 | 15.90 |
| Marto_VLP | ✓ | 52.11 | 50.57 | 37.16 | 35.71 | 25.72 | 22.37 | 22.96 | 13.20 | 16.90 | 20.19 |
| Marto_trel-image | ✓ | 51.64 | 50.28 | 38.01 | 34.60 | 27.15 | 22.66 | 21.84 | 12.77 | 14.80 | 19.06 |
| Marto_random-image | ✓ | 35.50 | 33.85 | 31.70 | 29.20 | 18.76 | 18.91 | 19.15 | 5.67 | 12.63 | 13.40 |
| Marto_oracle-best-image | ✓ | 62.50 | 60.12 | 44.20 | 41.21 | 38.65 | 28.91 | 25.03 | 19.78 | 23.81 | 24.35 |

vs. Marto_w/o QC). We also record the percentage of topics and questions that benefit from the addition of images. We find that the retrieval performance of nearly 80% of questions gets improved by adding the relevant images, covering around 90% topics. This finding confirms that while the majority of questions benefit from image attachment at the model level, some do not. Thus, predicting the suitability of questions for image attachment is necessary.

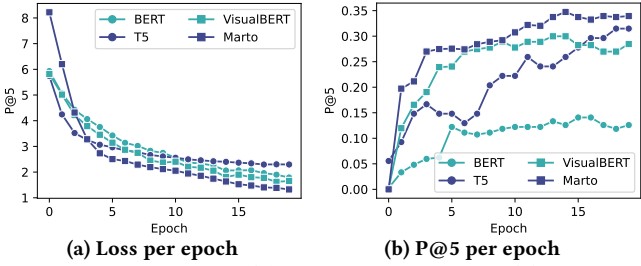

**(a) Loss per epoch**  **(b) P@5 per epoch**

**Figure 6: Training loss (a) and validation P@5 (b) score at different training epochs.**

**Impact of image variations (RQ2).** To further investigate the impact of images, we adopt several variants: Marto_random-image, Marto_trel-image, and Marto_oracle-best-image. We observe that even including a random image helps improve the retrieval process. Our findings further indicate that selecting and attaching the right image has a strong impact on the performance. Compared with attaching the random image (Marto_random-image), choosing the topic related image (Marto_trel-image) demonstrates a substantial improvement in performance, with a 45% increase in MRR rate. Overall, using the CLIP model for selecting question-relevant images in Marto shows further advantage, verifying the effectiveness of our image selection module. We also find that adopting VLP as Marto's base model decreases the performance, showing the effectiveness of using VLT5 as the base model. Overall, Marto demonstrates superior performance compared to another multimodal models and variants across multiple evaluation metrics.

**Generative vs. discriminative modeling (RQ3).** Table 3 shows that in both unimodal and multimodal scenarios, compared with

**Table 4: Comparison of the memory usage, the number of model parameters, and training inference time per epoch.**

| Model | Memory | Parameters | Train. time | Inf. time |
|---|---|---|---|---|
| BERT | 5621M | 110M | 7.30 min | 0.82 min |
| T5 | 8753M | 220M | 0.48 min | 0.33 min |
| VisualBERT | 7189M | 110M | 8.26 min | 1.13 min |
| Marto | 9965M | 220M | 0.77 min | 0.67 min |

**Table 5: Comparison of Marto's performance under Melon subset where all the questions originate from ClariQ.**

| Ans src. | Img. | MRR | P@1 | P@5 | nDCG@1 | nDCG@5 |
|---|---|---|---|---|---|---|
| ClariQ | ✗ | 30.95 | 28.57 | 29.52 | 14.40 | 14.19 |
| ClariQ | ✓ | 39.77 | 36.46 | 32.50 | 20.73 | 18.16 |
| Melon | ✗ | 28.85 | 30.95 | 28.10 | 19.00 | 15.79 |
| Melon | ✓ | **54.81*** | **53.73*** | **40.18*** | **30.98*** | **23.90*** |

discriminative retrieval models (e.g., BERT, VisualBERT), generative retrieval methods (e.g., T5, Marto) improve overall performance. This indicates that generative models are more effective in incorporating and unifying multimodal (image + text) information. To further address **RQ3** on the efficiency of generative retrieval models, we conduct experiments and analyze the training progress in terms of loss and validation P@5 score (Fig. 6). We compare the efficiency of these models and report the training and inference time of each epoch in Table 4. As shown in Fig. 6, we observe that: (i) generative models (dark blue) outperform discriminative models (light blue) in both performance and training efficiency. Despite having more parameters than VisualBERT, Marto requires significantly less training time (0.77 vs. 8.26) and inference time (0.67 vs. 1.13). (ii) Marto achieves the best overall convergence speed and the best validation score with the shortest time among all baselines, indicating that it can easily adapt to the downstream retrieval task with the same learning objective as in the pre-training stage.

**Multimodal impact on user answers (RQ4).** To answer RQ4 *from the model perspective*, in Table 5 we report the document retrieval performance on a subset of our dataset. We utilize the subset of Melon questions originate from ClariQ (**set 1 questions**), and create

**Table 6: Case study on the human-labeled multimodal questions. The up arrow means performance increases after adding images (VEQ), while the down arrow means performance decreases after adding images (TEQ).**

| Idx | Category | Topic | Facet | Clarification question | | Answer | Pred |
|---|---|---|---|---|---|---|---|
| 1 | Shopping | bowflex power pro | Find information about the Bowflex Power Pro. | Do you want to buy some parts for this equipment? | | Yes, I'm interested in what the material is. | ↑ |
| 2 | Location-related | map of Brazil | I am looking for information about taking a vacation trip to Brazil. | Would you like to see a specific region of Brazil? | | Yes. The pyramid in the picture looks interesting. | ↑ |
| 3 | General | titan | Find the homepage for Titan motorcycles. | Are you interested in a specific titan? | | No, I'm not interested in the anime, but the motorcycle. | ↑ |
| 4 | Recipe-related | salads | Find salads that are both nutritious and vegetarian-friendly. | Would you like to know about the recipe of this? | | Yes, it looks good. | ↓ |
| 5 | Categorical | insects | Find information on different types of insects. | Here's one example, is that what you need? | | Yes, exactly. | ↓ |

distinct variants by combining them with various answer sources. *A src.* column specifies where the user's answer originates from. The outcomes indicate a significant improvement in model performance by replacing the ClariQ answers with Melon answers (2nd row vs. 4th row). This indicates that users can offer more comprehensive and expansive answers when provided with multimodal questions, as the answers derived from Melon are closely aligned with the corresponding images. These results highlight the advantages of collecting new answers, aligning with the findings in Section 4.2. Significantly, we observe that multimodal information improves the results both in ClariQ and Melon answer sets (1st row vs. 2nd row; 3rd row vs. 4th row), which suggests that the images serve as a highly informative resource for the system to address the underlying user intent, aligning with findings for **RQ1**.

**Impact of the number of images**. In order to assess the impact of the number of attached images on the performance of Marto and VisualBERT, we compare their P@5 performance, as depicted in Fig. 7. Our observations reveal that incorporating images into the clarifying questions enhances the performance of both models compared to using text-only questions (image count 0). Furthermore, we find that attaching the top-one image yields the best performance, surpassing the results obtained when attaching 2 or 3 images. These findings validate the effectiveness of the image selection module in our model, emphasizing the importance of selecting the most relevant image to enhance overall performance.

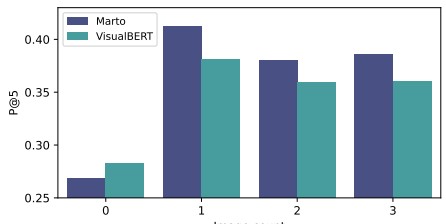

**Figure 7: The oracle P@5 retrieval performance concerning the number of images attached to the question.**

**Case study**. To illustrate the correlation between human judgment and model prediction, we present a set of human-assessed multimodal questions and provide the corresponding model's prediction; see Table 6. In most cases, including images in clarifying questions can provide valuable extra information (e.g., case 1, 2). Notably, in some cases, the clarifying question may be focused on a different aspect than the underlying user intent (e.g., case 3). However, the negative feedback received from users after viewing the image can still be valuable in improving document retrieval. However, there are some cases where images can be misleading. One such case is when the facets contain specific restrictions that are hard for the system to notice. For example, in case 4, the user intends to provide a clarifying question for a vegetarian-friendly recipe. Unfortunately, unbeknownst to the user, the image provided includes ingredients like shrimp that contradict the hidden facet of vegetarian-friendly recipes. Since the system relies on visual cues, it may prioritize suggestions based on the image's appearance, resulting in non-vegetarian salad recipes being offered. Another case where images can introduce bias is in case 5. The system may misinterpret the image as an indication of the user's specific interest in butterfly species. Consequently, it could prioritize retrieving information related only to butterflies rather than diverse insect species.

## 7 CONCLUSION

We investigate the novel task of asking multimodal query clarification in a text-dominant conversation. To provide an offline evaluation method for this system, we collect a dataset named Melon, with over 4k multimodal clarifying questions and 14k corresponding images. We also propose a multimodal query clarification model named Marto which adopts a prompt-based generative fine-tuning strategy for different subtasks. Experiments show that adding images can help improve retrieval results and lead to a significant performance lift. Additionally, multimodal contents result in more contextualized answers with richer semantic information. Compared to discriminative models, Marto demonstrates superiority in the document retrieval task. For future research, we list some ongoing directions in Appendix E.

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

## A  MELON QUALITY CONTROL

We implement several quality control measures. Firstly, we utilize the quality control mechanisms provided by Appen and AMT. We require AMT workers to have at least 10,000 approved HITs and a lifetime approval rate greater than 97%. We administer an onboarding test to all annotators to ensure their understanding of the task. After **Phase 1** and **Phase 2**, two quality checkers are employed and instructed to evaluate the quality of the returned questions and images by considering three criteria: (i) relevance of the clarifying question to the topic, (ii) suitability of the clarifying question to be accompanied by images, and (iii) relevance of the images to the clarifying question. We require that the clarifying questions satisfy all three criteria to be included in the dataset; 2.1% of the questions are marked as erroneous or invalid and are removed from our dataset. For answer submissions, we conduct manual checks by randomly examining 10% of the submissions.

To ensure the overall quality of the data collection stages, a subset of 200 clarifying questions, along with their corresponding information need, facet, and images, are sampled. These samples are then independently assessed by three crowdworkers. The evaluation focuses on determining the suitability of the questions for image attachment and the relevance of the attached images to the user's information need. The results indicate that 98% of the questions were judged as suitable for image attachment, while 96% of the images were found to be relevant to the user's information need. The small margin of difference between the suitability of questions for image attachment and the relevance of the attached images to the information need can be attributed to the explicit instructions

**Table 7: Top-8 key phrases in set 1 & 2 questions.**

| | |
|---|---|
| Set 1 | know, looking for, want, interested in, specific, information, referring to, see |
| Set 2 | want, see, interested in, looking for, know, information, see photos, different |

given to the annotators to provide diverse images. While this might have led to some images being topically relevant but not directly aligned with the user's specific information need, the small margin indicates overall, the dataset collected is of good quality.

## B  MELON QUESTION TERM COMPARISON

To examine potential biases in the collection of multimodal questions, we conduct a comparative analysis of the language usage in ClariQ questions (**set 1 questions**) and AMT questions (**set 2 questions**). Table 7 presents the top-8 key phrases extracted from both sets. There is a notable overlap in the distribution of key phrases between the two sets with some words (e.g., "want," "know") appearing frequently in both cases. Both sets of questions exhibit references to vision-related aspects, as indicated by the presence of terms like "see." Set 2 questions demonstrate a higher inclination towards vision-related phrases, such as "would you like to see" or "do you want photos," suggesting a stronger focus on image-related inquiries.

## C  EXPERIMENTAL SETUP

**Dataset..** Following [, Aliannejadi2020ConvAI3GC] we split our dataset into training/validation/test sets on different facets with 80%/10%/10% proportions, resulting in 856/107/107 facets in each set. The training, validation, and test set contain 14,187/1,851/1,865 samples, respectively.

**Evaluation metrics and statistical test**. For document retrieval, following [4, 20], we adopt mean reciprocal rank (MRR), precision (P@k), normalized discounted cumulative gain (nDCG@k), expected reciprocal rank (ERR@k) where $k \in \{1, 3, 5\}$ as evaluation metrics. The ground-truth relevance documents are obtained from TREC and adjusted on the facet level following [4]. We perform statistical significance testing using a two-tailed paired t-test with Bonferroni correction at a 99.9% confidence interval ($p < 0.001$).

**Compared methods**. To demonstrate the effectiveness of multimodal content in query clarification, we adopt various competitive baselines. We first consider several lexical methods:

- **OriginalQuery** [4] reports the performance where retrieval is performed only on the user query without clarification.
- **QL** [4, 46] is a query likelihood (QL) retrieval model that assigns different weights to the query, clarifying question and answer.
- **BM25** [53] is used to directly retrieve and rank the documents given the query, clarifying question, and answer.
- **LambdaMART** [67] is a learning to rank (LTR) baseline that learns to rank the documents according to queries. We use the 46 features listed in [47].

We then adopt the following pipeline-based methods:

- **BERT+CLIP+CLIP** adopts the BERT model to perform question classification and the CLIP model for the image selection. For document retrieval, we also utilize CLIP retrieval for encoding all images and questions, as well as all document identifiers.

- **BERT+CLIP+VLT5** utilizes the BERT model to perform question classification, CLIP model for image selection and the VLT5 model for document retrieval.

Next, we adopt baselines also under a multi-task framework with the first two in unimodal and the rest in multimodal scenario. For unimodal baselines, we adopt BERT and T5 respectively as the base model to retrieve the documents according to the topic and text-only clarifying question:

- **BERT** [24] shows the performance under the unimodal BERT ranking model, where clarifying questions are text-only. We use the BERT ranking implementation released in [39].
- **T5** [50] adopts a generative retrieval setting to retrieve the documents by the text-only questions. The generative model is trained to generate the keyword sequence of the relevant documents and can be seen as the text-only version of Marto.
- **VisualBERT** is the MQC pipeline with the same training tasks as Marto but based on VisualBERT model.
- **VisualBERT_w/o QC** [34] takes the query, the clarifying question with images, and the answer as input, without performing multimodal-enhanced question classification.

Finally, we compare Marto with multiple variants to evaluate the efficacy of our model's design.

- **Marto** is our model described in Section 5.
- **Marto_w/o QC** is a variant of our model where the multimodal-enhanced question classification module is removed.
- **Marto_VLP** [77] uses another state-of-the-art multimodal generative pre-trained model VLP as base model rather than VLT5.
- **Marto_random-image** is another Marto variant where we randomly attach an image from image pool in the retrieval process.
- **Marto_trel-image** is a Marto variant where we attach the topic-relevant images rather than question-relevant images.
- **Marto_oracle-best-image** is an oracle variant of our model where the model always selects the best image.

**Hyperparameter settings**. Our code is based on Pytorch [45] and Huggingface Transformers [66]. For VLT5, we use the pre-trained base version. By default, we set the batch size to 32 and the learning rate to 5e-5, the model is fine-tuned for 5 runs with 30 epochs per run, each with different random seeds. In the test stage, all models decode words with beam size 15. For first-stage document retrieval, we use BM25 [53] to retrieve 100 documents for each facet. We only report the result on the second-stage re-ranking.

## D  ETHICS STATEMENT

When creating the dataset, we take stringent measures to ensure the confidentiality of user intentions (facets) by keeping them strictly hidden from the annotators. Additionally, we prioritize the privacy of the collected images, implementing safeguards to protect sensitive and personal information. To assure that, we will only disclose the image URLs when sharing the dataset.

## E  ONGOING DIRECTIONS

The effectiveness of Marto indicates its potential for future directions, such as exploring the MQC task in a multi-turn scenario and improving image selection methods. Furthermore, it is intriguing to investigate how images can enhance the user experience at other

interaction stages beyond clarification. As image selection is critical, improving methods for selecting images and considering their diversity is vital. Therefore, it's of great research value to explore methods with a more sophisticated structure. Additionally, considering the increasing research interest in large language models (LLMs) for various NLP and IR tasks, we are extending current Marto (based on smaller-scaled pretrained model VLT5) to adopt multimodal LLMs as base model, such as BLIP-2 [33]. By comparing the upgraded Marto with existing LLMs such as GPT-4V [44], Llama [61], we are seeking to see how LLMs will help improve the clarification phase in the search process.

