# OpenReview forum: "Asking Multimodal Clarifying Questions in Mixed-Initiative Conversational Search"
_ACM.org/TheWebConf/2024/Conference — TheWebConf24 Oral_

### Official Review · Reviewer_wwYc · 2023-11-01

**Novelty:** 6
**Technical Quality:** 5

**Review:**

In this paper, the authors study the problem of multimodal clarification for conversational search. Their work is novel because it is the first to add images to conversational search clarification. The motivation that adding images to clarification as further contextual information and getting more accurate answers from users for retrieval is sensible. To advance research in this problem, the authors collect a new dataset called Melon and investigate four research questions in their experiments which all make perfect sense and form a good flow.

Pros:
1. The motivation of the work is well argued. The collecting pipeline of their dataset seems reasonable and results in a high agreement. I believe that their dataset has higher quality than the ClariQ dataset according to Table 2 and Figure 3. That could benefit the conversational search field. Overall, the paper is well-written and easy to follow.

2. With Figure 2, Section 3, and Section 5, the authors gave a clear description of their proposed multimodal clarification task and their proposed retrieval system with multimodal clarification, except for one answer function which I will describe below.

3. The research questions are interesting and their corresponding experiments are well designed. From Section 6 and Table 3, I am convinced that having images with textual clarification will benefit the final retrieval (RQ1), and having the right images is also important (RQ2).  Then, they show that generative models have both better effectiveness and efficiency (RQ3) in Figure 6 and Table 4, respectively. Table 5 shows that the retrieval system with responses to multimodal clarification significantly outperforms systems with responses to textual clarification. Further strengthening their numbers in Table 1, 2, and Figure 3. In conclusion, all their research questions are sufficiently investigated and argued.

4. The authors provide good case studies with both success and failure cases. They help understand what is going right and wrong with the multimodal clarification in reality.

Cons:
1. I notice that the authors define an answer function $A(t, f, q, I)$ in line 494. Based on my understanding, this function aims to simulate users' responses to the multimodal clarification from the search system, given $t,f,q,I$, especially $I$. However, there is no further mentions of this function $A$ or how it is implemented in the rest of the paper. From the motivation in Section 3, the quality of user responses is a key reason for multimodal clarification to work. Not properly implementing this could cause issues in analyzing the effect of image variations. (see my questions below)

2. Large language models have recently been shown highly effective in conversational search systems, such as generating clarification [e.g., 1] or simulating user responses to them [e.g., 2]. However, the authors have not discussed them in this work. In their experiments, the authors could compare their models with recent large language models such as GPT-4 or llama2 instead of BERT when possible.


Overall Evaluation:

References
1. Mulla, N., Gharpure, P. Automatic question generation: a review of methodologies, datasets, evaluation metrics, and applications. Prog Artif Intell 12, 1–32 (2023). https://doi.org/10.1007/s13748-023-00295-9
2. Paul Owoicho, Ivan Sekulic, Mohammad Aliannejadi, Jeffrey Dalton, and Fabio Crestani. 2023. Exploiting Simulated User Feedback for Conversational Search: Ranking, Rewriting, and Beyond. In Proceedings of the 46th International ACM SIGIR Conference on Research and Development in Information Retrieval (SIGIR '23). Association for Computing Machinery, New York, NY, USA, 632–642. https://doi.org/10.1145/3539618.3591683

**Questions:**

According to the cons listed above, my questions are:

1. Is this function $A$ implemented as part of the Marto model or replaced by the collected answers from MTurk? (From Figure.4, I assume the latter is true)

2. If the latter is true, is it accounted for in the RQ2 experiment, which means the MTurk annotators should generate answers given random/trel/oracle-best images. I don't question that their conclusions about RQ2 will remain the same either way, but it should be the correct way for analyzing the effects of various images.

**Ethics Review Description:**

No ethics issue noticed

**Reviewer Confidence:**

3: The reviewer is confident but not certain that the evaluation is correct

**Scope:**

4: The work is relevant to the Web and to the track, and is of broad interest to the community

---

### Official Review · Reviewer_JQBh · 2023-11-19

**Novelty:** 4
**Technical Quality:** 4

**Review:**

#### [Summary]
This paper targets query clarification in conversational search problems. The main motivation is that the addition of relevant images can help to clarify questions within a conversational context. The major contributions of this paper are: (1) providing a new dataset called Melon for multi-modal query clarification in conversational search, and (2) proposing a new multi-modal query clarification model called Marto. For the Melon dataset, the annotators for dataset creation are recruited from Appen, and the data collection processes are described in detail. According to the paper, the proposed dataset will be released upon acceptance. While the details and rationality of the proposed datasets are well provided, the proposed method has several weaknesses and needs more justification for its design.


#### [Strengths]
- This paper is well-written and clearly organized. The overall presentation quality is good.
- This paper introduces a new dataset for multi-modal query clarification in conversational search. The details of the dataset construction processes are well described.
- This paper provides extensive experimental results with a statistically significance test.

#### [Weaknesses]
- There is no explicit design to assess the relevance (or quality) of the selected image in Marto. After deciding whether to add an image or not, Marto simply uses the retrieved image (Eq.2) without any verification step. Considering Eq.1, the choice of adding an image depends on the quality of the user query. However, the low quality of the query does not necessarily mean that the retrieved image is helpful. The retrieved image may also have low quality and may not align with the user intent, which can amplify the incorrect understanding of the model. This is a very important factor when we consider adding information from a new modality. I believe this aspect should have been considered in this paper, as it argues the importance of using new modality information.

- The rationale of generative retrieval in Marto is unclear. I noticed that several retrieval approaches are employed simultaneously in Marto. For image selection, Marto uses cross-modal retrieval based on the encoded embeddings from CLIP, which follows the typical dense retrieval approach. For document retrieval, Marto follows the generative retrieval approach which directly generates the document identifiers. I expected more specific and plausible justifications for these choices, such as why they might be particularly beneficial for this problem. However, the author merely mentioned that the generative approach shows great research potential.

**Questions:**

I don't have any concerns about the dataset generation process. Nevertheless, I do have some concerns regarding Marto, specifically (1) the absence of a design to assess the quality of retrieved images and (2) the rationale behind the generative retrieval. Please refer to the weaknesses described in my review. Thank you.

**Ethics Review Description:**

I don't have ethical concerns

**Reviewer Confidence:**

3: The reviewer is confident but not certain that the evaluation is correct

**Scope:**

4: The work is relevant to the Web and to the track, and is of broad interest to the community

---

### Official Review · Reviewer_2d6N · 2023-11-23

**Novelty:** 5
**Technical Quality:** 5

**Review:**

This paper proposes a task of multimodal clarifying questions for conversational search. It creates a dataset named Melon that enriches the ClariQ data with images and more manually written clarifying questions. It also proposes a multimodal query clarification model that retrieves clarifying questions, classifies whether the question should be assisted with images, retrieves images if yes, and conducts document retrieval based on the previous conversation context. The authors have made a lot of effort on the dataset construction and each step in the workflow. Overall, I lean towards "weak accept”, but also have concerns.

The pros:
1. The writing is good mostly.
2. The constructed dataset can encourage future research in this direction.
3. It makes sense to classify whether images need be be retrieved for a clarifying question.

The cons:
1. This work focuses on document retrieval based on mixed-initiative conversations, but why not answer the questions directly instead of retrieving documents? The example shown in Figure 1 also gives a solution instead of showing documents. In the era of LLMs, returning the answers summarized from the documents is more promising compared to showing document lists.
2. Similarly, a clarifying question is also chosen from a predefined collection, which is again not very promising, especially when the modern LLMs have strong abilities of generation.
3. The choice of each model in the workflow is not justified, e.g., why does the final document retrieval use doc-id-based generative retrieval? Why does the question generation model use VLT5 as the backbone while the image retrieval uses the CLIP?
4. Several other questions:
* a. Do questions of set 1 and set 2 have topic overlap?
* b. What are the features used in LambdaMart?
* c. How are the keywords used as the document ID extracted?
* d. Why do the document retrieval metrics only focus on top results?
* e. Line 178: Why only mention efficiency? What about effectiveness?

**Questions:**

Please see the overall review.

**Reviewer Confidence:**

3: The reviewer is confident but not certain that the evaluation is correct

**Scope:**

4: The work is relevant to the Web and to the track, and is of broad interest to the community

---

### Official Review · Reviewer_Qyn3 · 2023-11-26

**Novelty:** 6
**Technical Quality:** 5

**Review:**

This paper is concerned with the use of clarifying images in a search clarification scenario. A dataset (based on the existing Clariq dataset) is created and a model are proposed.

I like the envisaged task - the opening example is convincing.

I dislike the phrasing of RQ4 given that its still an analysis of the test collection - its not a user study per se. Claiming you can impact on the user (response) is therefore a bit of an overclaim in my opinion.

As I'm not 100% familiar with Clariq, I'm confused what document corpus is being retrieved from. You have a LambdaMART baseline, and the feature definitions cite the LETOR 4 paper, so I'm inclined to think this is all based on LETOR 4. Pleae clarify?

The use of a generative model for document retrieval confuses me. It seems unrelated to the contribution at hand. We know they don't currently scale to many documents. Why not use a more conventional dense bi-encoder here? As your dataset is LETOR, are you actually using it as a re-ranker?

Figure 6(a) appears to show that Precision @5 goes DOWN per epoch. Please check the ylabel? I suspect it should be loss.

The experimental setup is in the Appendix. Even after checking it, I would have liked more information about how exactly the system is evaluated. Section 3 defined a 5 step workflow. Is evaluation only conducted based on retrieval performance? How do we know that images clarifications being retrieved for inappropriate questions - c.f. step (iv). I would have expected some classification evaluation for that?

Minor comments:
 - line 242: state-of-the-art in what?
 - line 318: 'presents relevant documents to the user' - are you sure they are relevant? Its retrieved, right?
 - Figure 3: please clarify in the caption that answer length wrt terms - otherwise, we dont know what is being counted - its not directly clear when the figure is introduced around line 449

**Questions:**

- How did you choose the examples in the Case Study?
- Underlying document corpus?
- Generative vs dense retrieval?
- How does evaluation correspond to the steps in Section 3.
- reusability - can the dataset be used by future improved systems?

**Reviewer Confidence:**

3: The reviewer is confident but not certain that the evaluation is correct

**Scope:**

4: The work is relevant to the Web and to the track, and is of broad interest to the community

---

### Official Review · Reviewer_GMED · 2023-11-29

**Novelty:** 5
**Technical Quality:** 6

**Review:**

The paper introduces a novel task in open-domain mixed-initiative conversational search systems, focusing on using multimodal clarifying questions that integrate image data to improve document retrieval. Key points include:

1. **Novel Task Introduction**: Integration of text and image data in conversational search queries.
2. **Dataset and Benchmarking**: Development and use of a large-scale dataset named Melon, benchmarked with a multimodal query clarification model, Marto.
3. **Impact Analysis**: Examination of how multimodal questions affect user responses and retrieval performance.
4. **Training and Comparative Analysis**: Detailed multi-staged training process for the system, with significant improvements in document retrieval demonstrated through comparisons with baseline models.

The strengths and weaknesses of the paper can be summarized as follows:

**Strengths**:
1. **Improvement in Retrieval Performance**: The paper effectively demonstrates how Marto's retrieval performance significantly benefits from incorporating multimodal clarifying questions in a mixed-initiative conversational search system.

**Weaknesses**:
1. **Methodological Concerns in Question Classification**:
   - The 'Multimodal-enhanced question classification' section (Line 520) presents a methodological approach that classifies clarifying questions in Melon into Textual-Enhanced Questions (TEQ) or Visual-Enhanced Questions (VEQ) based on their suitability for image attachments post-annotation.
   - This classification appears more as a filter to remove unsuitable training data, particularly questions with inadequate image pairings, rather than a robust methodological feature.

2. **Evidence of Data Malformation**:
   - Examples 4 and 5 in the Case Study section (Line 901) suggest instances of data malformation, where inappropriate image selections by annotators led to unqualified training instances.

3. **Issues with Data Preparation**:
   - The strategy to classify and filter questions may address poorly formed training examples or data noise, but it could be more effective if these issues were handled during the dataset collection phase or before entering the experimental pipeline.
   - This raises questions about the necessity of 'predicting the suitability of questions for image attachment' (Line 720), suggesting that the core problem lies in initial data preparation rather than in the system's inherent capabilities.

**Questions:**

Sure, here are your queries in a more concise form:

1. **Typographical Error in Line 504**:
   - Is the sentence in Line 504 a typo? Should it read, “when a conversational system enters the clarification question phase after a user query submission”?

2. **Annotated Unimodal Clarifying Questions (Line 355)**:
   - Are the unimodal clarifying questions referenced in Line 355 removed from the dataset, as suggested by the statement in Line 366?

3. **Notation Clarification for VLT5 (Line 546)**:
   - In the expression et = ΦT (<p1>, t, q), what do 't' and 'q' represent? Is there a clearer explanation for these symbols?

**Reviewer Confidence:**

3: The reviewer is confident but not certain that the evaluation is correct

**Scope:**

4: The work is relevant to the Web and to the track, and is of broad interest to the community

---

### Decision · Program_Chairs · 2024-01-22

**Decision:**

Accept (Oral)

**Comment:**

In this paper, the authors investigate the multimodel clarification question generation problem.
 Reviewers agree that the paper has the following pros:
 (1) the presentation of the paper is generally good.
 (2) the dataset constructed in the paper is valuable.
 (3) experimental results are significant.

 Reviewers also found several cons of the paper:
 (1) motivation of some design choices. It would be great if the authors could pay more attention to the motivation and justification of some specific design choices made in the paper, for example, why document retrieval is used, why LLMs are not used, why a generative model is used, etc.